# Ubiquitin and a charged loop regulate the ubiquitin E3 ligase activity of Ark2C

Andrej Paluda [1,2], Adam J. Middleton [1], Claudia Rossig [1], Peter D. Mace [1] & Catherine L. Day [1✉]

A large family of E3 ligases that contain both substrate recruitment and RING domains confer specificity within the ubiquitylation cascade. Regulation of RING E3s depends on modulating their ability to stabilise the RING bound E2~ubiquitin conjugate in the activated (or closed) conformation. Here we report the structure of the Ark2C RING bound to both a regulatory ubiquitin molecule and an activated E2~ubiquitin conjugate. The structure shows that the RING domain and non-covalently bound ubiquitin molecule together make contacts that stabilise the activated conformation of the conjugate, revealing why ubiquitin is a key regulator of Ark2C activity. We also identify a charged loop N-terminal to the RING domain that enhances activity by interacting with both the regulatory ubiquitin and ubiquitin conjugated to the E2. In addition, the structure suggests how Lys48-linked ubiquitin chains might be assembled by Ark2C and UbcH5b. Together this study identifies features common to RING E3s, as well elements that are unique to Ark2C and related E3s, which enhance assembly of ubiquitin chains.

[1] Biochemistry Department, School of Biomedical Sciences, University of Otago, Dunedin 9054, New Zealand. [2] Present address: TMDU Advanced Research Institute, Tokyo Medical and Dental University, 1-5-45 Yushima, Bunkyo-ku, Tokyo 113-8510, Japan. ✉email: catherine.day@otago.ac.nz

Attachment of ubiquitin to proteins is a key mechanism that determines the timing and strength of many cellular responses[1]. This is because the addition of ubiquitin (ubiquitylation) controls protein longevity, abundance, and function[2]. Ubiquitylation relies on the activities of an ATP-dependent ubiquitin-activating enzyme (E1), a ubiquitin-conjugating enzyme (E2) and a ubiquitin ligase (E3). While the concerted activity of all three proteins is required for modification of substrate proteins with ubiquitin, the E3 has a key role in determining both the timing of substrate ubiquitylation and the type of modification attached[3,4]. As a result, efforts to identify factors that regulate substrate ubiquitylation have focused on the analysis of the E3 ligases[5].

Most E3s possess a RING (Really Interesting New Gene) domain that binds the thioester linked E2~ubiquitin (E2~Ub) conjugate and brings it into close proximity with the substrate[3,6]. ubiquitin is then typically transferred directly from the E2 to a lysine residue in the substrate. Activation of transfer by the E3 relies on the ability of the RING domain, together with adjacent residues, to bind the E2 conjugated ubiquitin (referred to as the donor ubiquitin, or $Ub^D$) and restrict its position (Fig. 1a). The RING–E2 interaction is highly conserved[7], but E3–$Ub^D$ interactions are more variable and include conserved features as well as unique features such as phosphorylated residues distant to the RING domain;[8] extensions to the RING domain;[9] or for many E3s, a dimeric partner RING domain[10,11]. Irrespective of the basis of the contacts, interactions with $Ub^D$ is essential to stabilise the E2~Ub conjugate in the 'activated conformation' where the C-terminal tail of ubiquitin is extended and the thioester is primed for attack by a lysine[3,12]. Interactions between the E3 and $Ub^D$, therefore, regulate ubiquitin transfer, and this has prompted a focus on identifying the features in E3 ligases that stabilise the activated conformation of RING bound E2~Ub conjugates.

Two related RING E3 ligases, Arkadia (RNF111) and Ark2C (RNF165) (Supplementary Fig. 1a), have central roles in the regulation of Transforming Growth Factor β (TGFβ) signalling[13,14]. Both E3s are positive regulators of TGFβ signalling because they mediate the attachment of degradative ubiquitin chains and subsequent destruction of negative regulators of the TGFβ pathway, such as SMAD7 and SnoN/Ski. Alteration in Arkadia expression is associated with cancer[15] and in mice, homozygous deletion of Arkadia is embryonic lethal[16,17]. Meanwhile, loss of Ark2C, which is only expressed in the nervous system, causes motor innervation defects that lead to the death of most pups prior to weaning[14]. Arkadia also has important roles in regulating the DNA damage response pathway as it promotes the addition of Lys63-linked chains to the xeroderma pigmentosum group C (XPC) protein[18,19], which enhances repair because it stimulates the release of XPC from damaged DNA. Furthermore, Arkadia is required for the ubiquitylation and degradation of PML (promyelocytic leukaemia) protein following the addition of arsenic to cells—the conventional treatment for PML[20]. In part, ubiquitylation of PML and XPC by Arkadia is regulated by the addition of SUMO chains. This is because Arkadia contains three SUMO interaction motifs that bind to polySUMO chains[21], and only when PML and XPC have been modified by SUMO are they targeted for ubiquitylation by Arkadia[22]. Despite the importance of Arkadia and Ark2C, a detailed understanding of the features that regulate ubiquitin transfer remains uncertain.

Previously we reported that the RING domains of Arkadia and Ark2C are bona fide ubiquitin-binding domains, and that docking of the RING domain onto a *regulatory* ubiquitin molecule ($Ub^R$, Fig. 1a) enhances ubiquitin transfer[23]. Biochemical studies suggested that $Ub^R$ was required to prime the thioester bond between UbcH5b and $Ub^D$ for nucleophilic attack by an acceptor lysine. However, the specific details of the activated complex formed were uncertain, and it was unclear if additional features beyond the RING domain were necessary for optimal activity. Here, we report the crystal structure of the activated complex that includes the RING domain of Ark2C bound to both $Ub^R$ and the UbcH5b~Ub conjugate. This structure reveals the contacts between Ark2C, $Ub^R$ and $Ub^D$ that stabilise the activated complex, explaining why ubiquitin transfer by Ark2C is significantly enhanced by ubiquitin. In addition, we dissect the function of a loop that precedes the core RING domain in Ark2C. A model of the Ark2C–UbcH5b complex in the process of assembling Lys48-linked ubiquitin chains is also proposed. Together, our results suggest that multiple elements in Ark2C work together to regulate ubiquitin transfer.

## Results

**The activated UbArk2C–UbcH5b~Ub complex.** Previously we established that binding of ubiquitin to Ark-like RINGs significantly increases their activity, and we characterised the complex formed between the RING domain and ubiquitin[23]. However, our efforts to obtain a structure of the E2~Ub conjugate bound complex in which the E2 conjugated ubiquitin molecule occupied the activated conformation proved unsuccessful. Guided by the structure of $Ub^R$ bound to the RING domain we prepared a fusion protein (referred to as UbArk2C) in which ubiquitin was linked to the N-terminus of the RING domain of Ark2C (residues 255–346) (Fig. 1b and Supplementary Fig. 1b). Ubiquitin transfer by UbArk2C was increased compared to Ark2C RING alone in both discharge and multi-turnover assays suggesting that ubiquitin occupies the $Ub^R$ binding site (Supplementary Fig. 1c, d). Furthermore, UbArk2C behaved as a monomer suggesting that an intramolecular interaction was favoured (Supplementary Fig. 1e). Crystals that diffracted to 2.5 Å were obtained and the structure was solved by molecular replacement, with residues 1–74 of ubiquitin positioned, as well as residues 259–274 and 288–344 of Ark2C (Supplementary Fig. 2a and Table 1). Comparison of the UbArk2C structure with previous structures of the Ark2C–RING bound to ubiquitin reveals a high degree of similarity (Supplementary Fig. 2b). As a result, the UbArk2C fusion construct was used for the structural studies described here.

To understand the molecular basis of activation of ubiquitin transfer by Ark2C we solved the structure of the UbArk2C fusion in complex with UbcH5b~Ub (Fig. 1c). Crystals of the complex that diffracted to 2.8 Å were obtained following purification of the stable UbArk2C–UbcH5b~Ub complex (Supplementary Fig. 3a), and the structure was solved by molecular replacement (Table 1). The density was well defined for most of the structure, and for UbArk2C residues 1–73 ($Ub^R$), as well as 258–274 and 289–341 of Ark2C, could be positioned, while for the conjugate residues 1–147 of UbcH5b and 3–76 for ubiquitin were resolved. In contrast to previous Ark2C complex structures, UbArk2C–UbcH5b~Ub clearly adopts the primed conformation, with E2–Ub contacts similar to those observed in previous activated RING–E2~Ub complexes (Supplementary Fig. 3b)[3]. Importantly, Ile44 of $Ub^D$ interacts with the crossover helix of the E2, while the Ile36 centred patch of $Ub^D$ contacts residues in the RING domain, including His314 and Ile333. Furthermore, the β1–β2 loop of $Ub^D$ adopts the 'loop-in' conformation and is oriented towards the RING domain as observed in other activated complexes (Supplementary Fig. 3b, c)[24].

Activation of Ark2C and stabilisation of the primed conformation appears to be highly dependent on $Ub^R$. Contacts between residues in β2 of $Ub^R$ (Thr9, Lys11 and Thr12) and the C-terminus of α1 of $Ub^D$ (Gln31, Asp32 and Gly35) extend the E2~Ub binding surface by ~190 Å$^2$ so that in total ~1080 Å$^2$ of

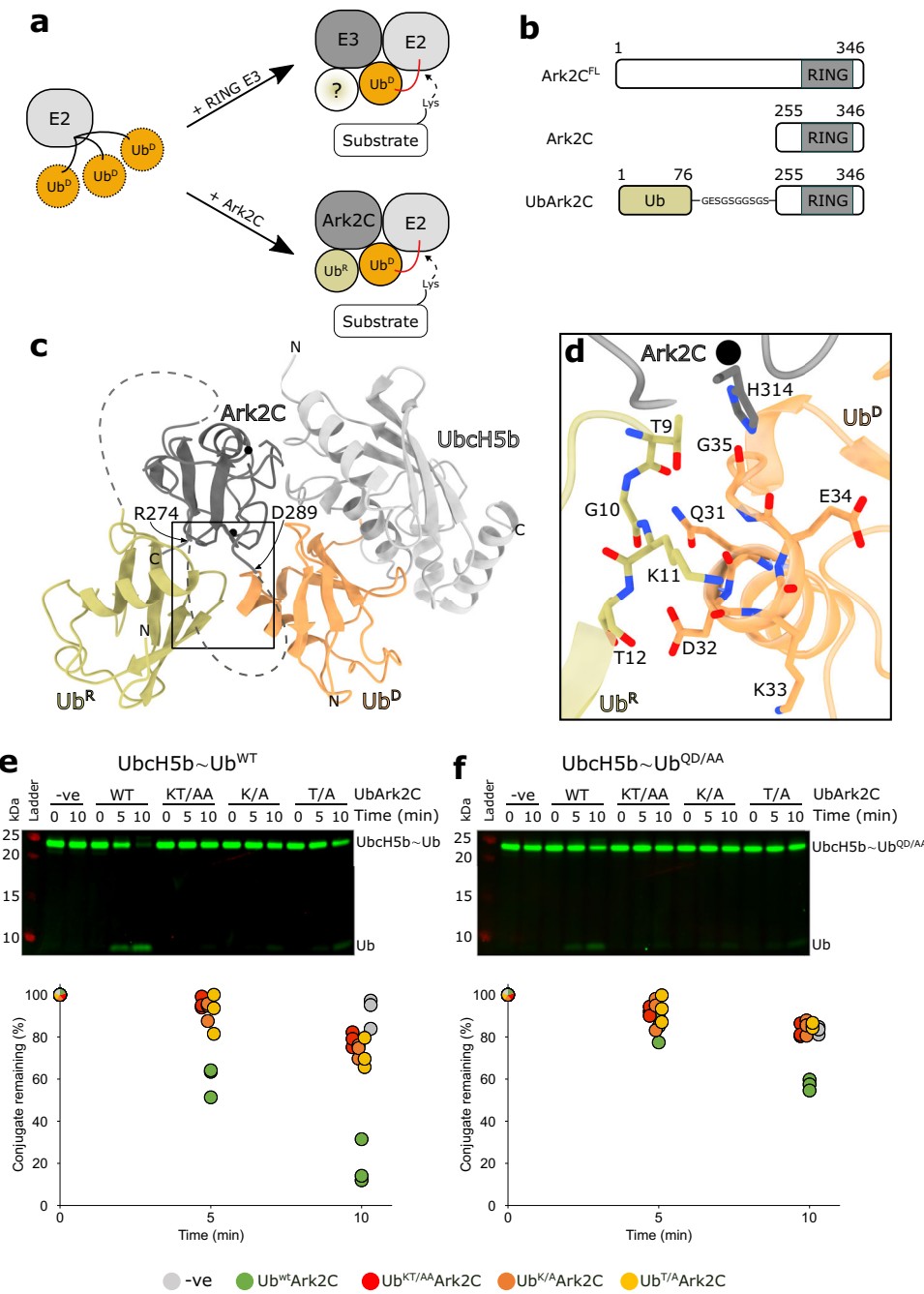

**Fig. 1 Structure of the Ark2C catalytic complex and analysis of the Ub$^R$–Ub$^D$ interface. a** Cartoon illustrating how RING E3 ligases stabilise the closed conformation of the E2-Ub conjugate (top) and the role of Ub$^R$ for Ark-like RINGs. **b** Schematic showing the domain architecture of Ark2C and the constructs used in this study. **c** Cartoon representation of the activated complex of UbArk2C–UbcH5b-Ub; Ark2C (dark grey), UbcH5b (pale grey), Ub$^R$ (yellow) and Ub$^D$ (orange). Black spheres depict zinc ions. **d** Close-up showing the contacts between Ub$^R$ and Ub$^D$ as sticks. Red and blue represent oxygen or nitrogen atoms, respectively. Carbon atoms are coloured as in (**c**). **e** Discharge assays assessing the ability of the indicated UbArk2C proteins to hydrolyse the WT UbcH5b-Ub conjugate. **f** Discharge assays as in (**e**) except Gln31 and Asp32 in ubiquitin conjugated to UbcH5b were mutated to Ala, referred to as UbcH5b-Ub$^{QD/AA}$. In both (**e**) and (**f**) ubiquitin used to prepare the UbcH5b-Ub conjugate was labelled with Cy3 and all gels were imaged and quantified using Image Studio Lite (Li-COR Biosciences). Experiments were performed in triplicate using 0.25 µM E3 ligases and 0.125 mM L-lysine. All repeats were quantified and are shown. Source data are provided in the Source Data file.

the E2~Ub conjugate is buried in the primed conformation (Fig. 1c, d). Comparison of the structures of Ub$^R$ bound Ark2C in the presence and absence of the E2~Ub conjugate shows that close-packing at the Ub$^R$–Ub$^D$ interface is, in part, achieved by displacement of Ub$^R$ in the activated complex (Supplementary Fig. 3d). To investigate the importance of interactions between Ub$^D$ and Ub$^R$ we mutated residues at the Ub$^R$–Ub$^D$ interface and

assessed activity. To specifically mutate Ub$^R$, but not ubiquitin that occupies the Ub$^D$ binding site, the mutations were introduced into the UbArk2C fusion protein. We prepared UbArk2C proteins in which Lys11 or Thr12 were mutated to alanine (Ub$^{K/A}$ or Ub$^{T/A}$), as well as a version in which both residues were mutated (Ub$^{KT/AA}$). The activity was analysed using single-turnover assays monitoring ubiquitin release from

the UbcH5b~Ub conjugate following the addition of UbArk2C. As expected, Ark2C fused to WT ubiquitin resulted in rapid discharge of the WT UbcH5b~Ub conjugate, whereas mutation of the contact residues in Ub[R] significantly impeded activity (Fig. 1e) and the residual activity was more comparable to that of Ark2C alone (Supplementary Fig. 1d). We also prepared a UbcH5b conjugate with ubiquitin in which Gln31 and Asp32 were mutated to Ala (referred to as UbcH5b~Ub[QD/AA]). In equivalent assays, the release of ubiquitin from the UbcH5b~Ub[QD/AA] conjugate was delayed for both the WT and mutant forms of UbArk2C (Fig. 1f), with the mutants again more comparable to Ark2C alone (Supplementary Fig. 3e).

Together these studies reveal the details of the Ark2C complex required for activation of UbcH5b~Ub and establish the importance of specific contacts between Ub[R] and Ub[D] for activation of ubiquitin transfer by Ark2C.

**A model for the assembly of Lys48-linked chains.** To assemble ubiquitin chains an acceptor ubiquitin (Ub[A]), with an appropriately positioned lysine residue, must also interact with the activated RING-E2~Ub conjugate complex. The UbcH5 family preferentially assembles Lys11, Lys48 and Lys63 linked chains[25,26], but only Ub[A] poised for the assembly of Lys11 linked chains by UbcH5a has been characterised[25]. In the structure of the activated Ark2C complex, crystal packing contacts position a ubiquitin molecule close to the active site of UbcH5b, such that Lys6 and Lys48 are ~11–13 Å from the active site raising the possibility that a Ub[A] binding site has been captured (Fig. 2a). The potential Ub[A]-UbcH5b interface buries 375 Å² and includes Asp116, Arg125 and Lys128 on UbcH5b (Supplementary Fig. 4a,

b). When Arg125 on the E2, which interacts with Glu64 on ubiquitin, was mutated to Asp (UbcH5b[R125D]) assembly of ubiquitin chains was reduced when wild-type ubiquitin was used (Supplementary Fig. 4c). Furthermore, when ubiquitin containing only a single lysine, Lys48 (Ub[K48]), was used in the assays, formation of high molecular weight chains was significantly impeded for UbcH5b[R125D] (Fig. 2b). Adjustments will be required to bring Lys48 close to the catalytic cysteine, however, NMR shows that Lys48 is flexible[27]. When the repertoire of conformations from solution studies is included, it is apparent that Lys48 could assume a conformation that would further reduce the distance to the thioester bond to ~5 Å (Fig. 2c). Given the well-established ability of UbcH5 E2s to build Lys48-linked chains, but not Lys6-linked chains, it seems likely that this structure presents one way in which UbcH5b can position Ub[A] for the assembly of Lys48-linked ubiquitin chains.

**Multiple ubiquitin molecules cooperate to promote chain assembly by Ark2C.** While ubiquitin transfer from an E2 to a substrate is highly reliant on interaction with an E3 and positioning of Ub[D] as well as Ub[A] molecules, for some E2s other allosteric effectors are also important[28]. In the case of UbcH5b, activity is often significantly increased when ubiquitin binds to the β-sheet on a surface opposite the catalytic cysteine—often referred to as the ubiquitin backside binding site (or Ub[B])[29,30]. Recent studies suggest that the primary role of Ub[B] is to stabilise the RING-induced conformational changes in the UbcH5b conjugate, so that attachment of both the initial ubiquitin and chain formation is increased[3,31,32]. To investigate if Ub[B] regulated the activity of UbcH5b when paired with Ark2C, the Ser22 to

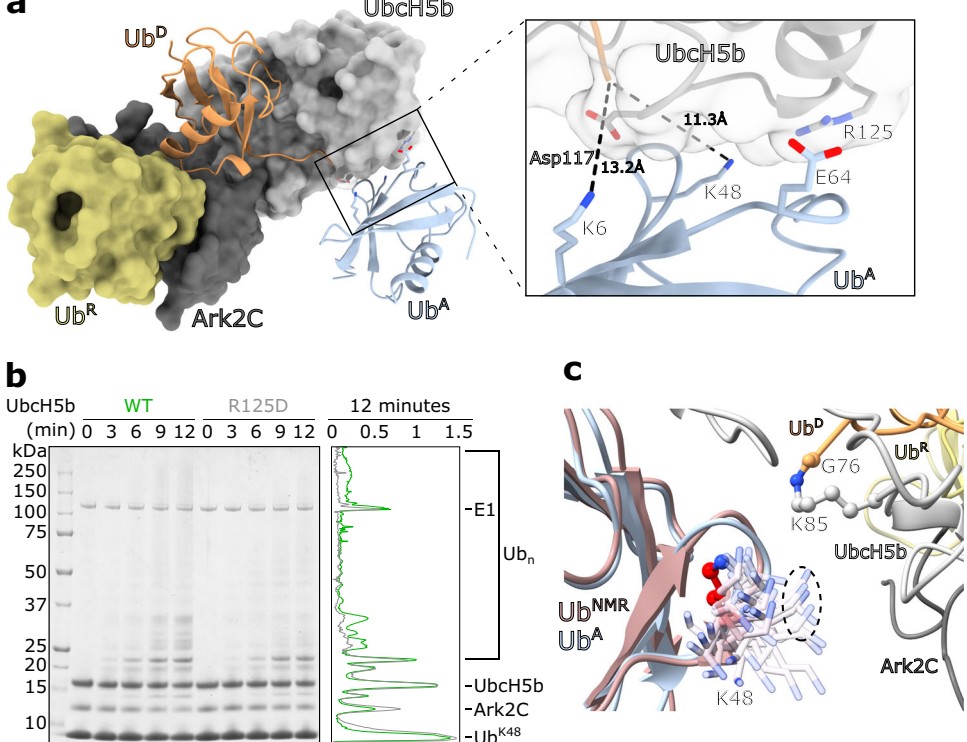

**Fig. 2 An acceptor ubiquitin binding site on UbcH5b. a** The UbArk2C-UbcH5b~Ub complex is shown as a surface except Ub[D] is shown as a cartoon. A potential Ub[A], which is the result of crystal packing is also shown as a cartoon (pale blue). The Lys residues that approach the active site and important residues in UbcH5b are shown as sticks (below). **b** Multiturnover activity assay comparing the ability of Ark2C to assemble Lys48-linked ubiquitin chains with UbcH5b WT and the UbcH5b R125D mutant. Lys48-only ubiquitin was included in the assays. A-line scan of the 12 min time point sample is shown at right (UbcH5b WT in green, UbcH5b R125D in grey). Intensity (arbitrary units) is plotted against distance migrated. **c** All 50 states of ubiquitin from the NMR structure (PDB: 2KN5) were overlayed onto Ub[A] from panel (**a**). The position of just the Lys48 sidechain in the ensemble is shown.

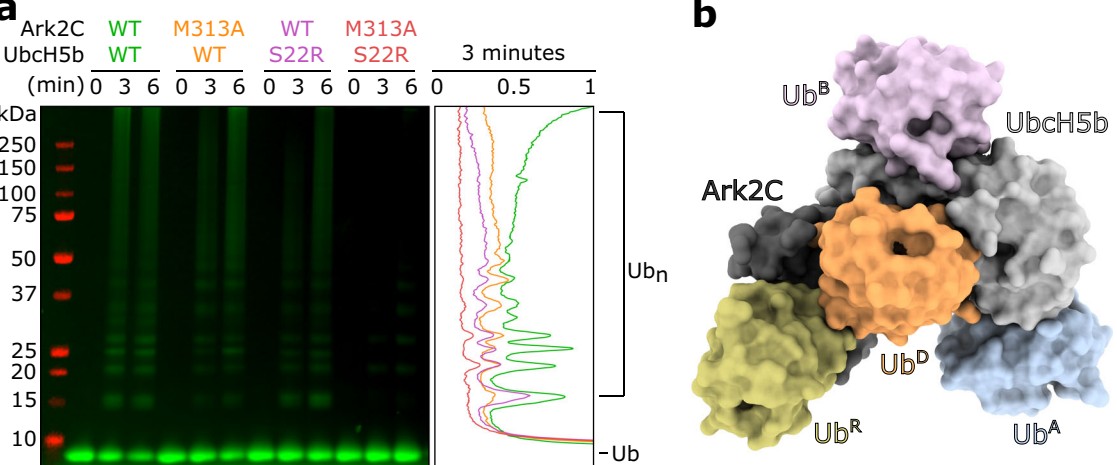

**Fig. 3 Multiple ubiquitin binding sites are needed for efficient transfer. a** Multiturnover assays comparing assembly of ubiquitin chains by Ark2C that can bind Ub$^R$ (WT) and Ark2C in which this interaction is disrupted (M313A). UbcH5b that retains the ability to bind Ub$^B$ (WT) and in which this interaction was disrupted (S22R) was compared. Cy3-labelled ubiquitin was used in the assay and the gel was imaged using fluorescence and Image Studio Lite. A linescan of the 3 min timepoint is shown at right. Specific E3–E2 combinations are colour-coded (Ark2C–UbcH5b: WT–WT in green, M313A–WT in yellow, WT–S22R in purple, and M313A–S22R in red). Intensity (arbitrary units) is plotted against distance migrated. **b** Model of the Ark2C ubiquitylation complex showing the position of ubiquitin molecules required for the efficient assembly of ubiquitin chains by Ark2C. Ub$^B$ from PDB: 4V3L was positioned by aligning the UbcH5b molecule onto UbcH5b in the activated Ark2C complex. All other molecules are from the structure of the activated complex.

arginine form of UbcH5b (UbcH5b$^{S22R}$), which prevents binding of Ub$^B$, was prepared[33]. In the absence of Ub$^B$ binding, efficient chain assembly by UbcH5b and Ark2C did not occur, with a significant reduction in high chains readily apparent (Fig. 3a). Introduction of the Met313 to Ala mutation in Ark2C (Ark2C$^{M313A}$) (Supplementary Fig. 2b), which blocks binding of Ub$^R$, also reduced activity, although not as much as when the Ub$^B$ binding site was mutated. However, when both ubiquitin binding sites were blocked by combining Ark2C$^{M313A}$ and UbcH5b$^{S22R}$ there was very little residual activity and even the formation of diubiquitin was impeded (Fig. 3a). These results indicate that both the Ub$^R$ and Ub$^B$ binding sites cooperate to enhance ubiquitin transfer by Ark2C.

Identification of a Ub$^A$ binding site on UbcH5b, as well as a demonstration of the importance of the Ub$^B$ binding for activity with Ark2C, allowed us to prepare a model of the Ark2C–UbcH5b catalytic complex that includes four ubiquitin molecules (Fig. 3b). Critically two ubiquitin molecules (Ub$^R$ and Ub$^B$) have regulatory roles, while the other two molecules (Ub$^D$ and Ub$^A$) are positioned ready for assembly of Lys48-linked diubiquitin.

**The β3-RING loop of Ark2C enhances activity.** Regulation of ubiquitin transfer by E3 ligases is tightly controlled and in many cases elements outside the RING domain have been found to help stabilise the activated conformation of the E2~Ub[3,5]. In Ark2C, the ends of a loop that connects the N-terminal β-strand (β3) to the core RING domain (referred to as the β3-RING loop; Fig. 4a and Supplementary Fig. 1a) are located adjacent to Ub$^R$ and Ub$^D$ (Fig. 4b), and we hypothesised that the loop may modulate the activity of Ark2C. The loop contains stretches of basic and acidic residues and is presumed to be flexible because it was not resolved in the UbArk2C structures reported here. Because the β3-RING loop contains a number of Lys residues and, Lys29 and Lys33 are exposed on both Ub$^R$ and Ub$^D$, we used an amine crosslinking approach to determine if the loop contacted ubiquitin. Incubation of the UbArk2C–UbcH5b~Ub complex with an amine crosslinker that had a spacer arm of ~11 Å resulted in a band corresponding to the mass of a ~1:1 complex (Supplementary Fig. 5a). Analysis

by mass spectrometry identified cross-links between Lys33 of ubiquitin and both Lys279 and Lys282 in the loop of Ark2C (Supplementary Fig. 5b). Although the crosslink could be between the β3-RING loop and either Ub$^D$ or Ub$^R$, this result suggested that the loop was in close proximity to at least one ubiquitin molecule.

To investigate if the β3-RING loop contributed to the activity we generated deletion constructs that were missing either the majority of the basic (Δ273–282) or acidic (Δ283–291) residues, referred to as Ark2C–ΔBasic and Ark2C–ΔAcidic, respectively (Fig. 4a). Although in multi-turnover assays the deletion mutants only showed a modest reduction in activity (Supplementary Fig. 5c), in single turnover assays both deletion constructs had a considerably diminished ability to promote ubiquitin release when incubated with the UbcH5b thioester conjugate (Fig. 4c). This suggests that the β3-RING loop promotes efficient ubiquitin transfer by Ark2C.

**Basic residues in the β3-RING loop contact Ub$^R$.** Given that the crosslinking data suggested that the β3-RING loop was close to at least one ubiquitin molecule, we wondered if the loop facilitated the recruitment of Ub$^R$. Initially, we used a GST-pulldown assay to evaluate ubiquitin binding to Ark2C (Fig. 4d). Like WT Ark2C, the Ark2C–ΔAcidic mutant pulled-down ubiquitin, whereas Ark2C–ΔBasic no longer interacted with ubiquitin. Analytical size exclusion chromatography (Supplementary Fig. 5d) supported this conclusion, with WT Ark2C coeluting with ubiquitin, but the Ark2C–ΔBasic mutant and ubiquitin eluted separately.

In the structure of the activated complex the four basic residues between positions 271 and 274 are in close proximity to Ub$^R$, with the sidechain of Arg273 close to the C-terminus of the α-helix in ubiquitin (Fig. 4e), and we focused our attention on these. We generated variants of Ark2C in which Lys271 and Lys272 were mutated to alanine (Ark2C$^{KK/AA}$); or Arg273 and Arg274 were mutated to alanine (Ark2C$^{RR/AA}$). In GST-pulldown assays, both mutants had a reduced ability to bind ubiquitin, with mutation of the two Arg residues significantly reducing ubiquitin binding by Ark2C (Fig. 4f). Furthermore, consistent with the requirement for Ub$^R$ to promote ubiquitin transfer by Ark2C,

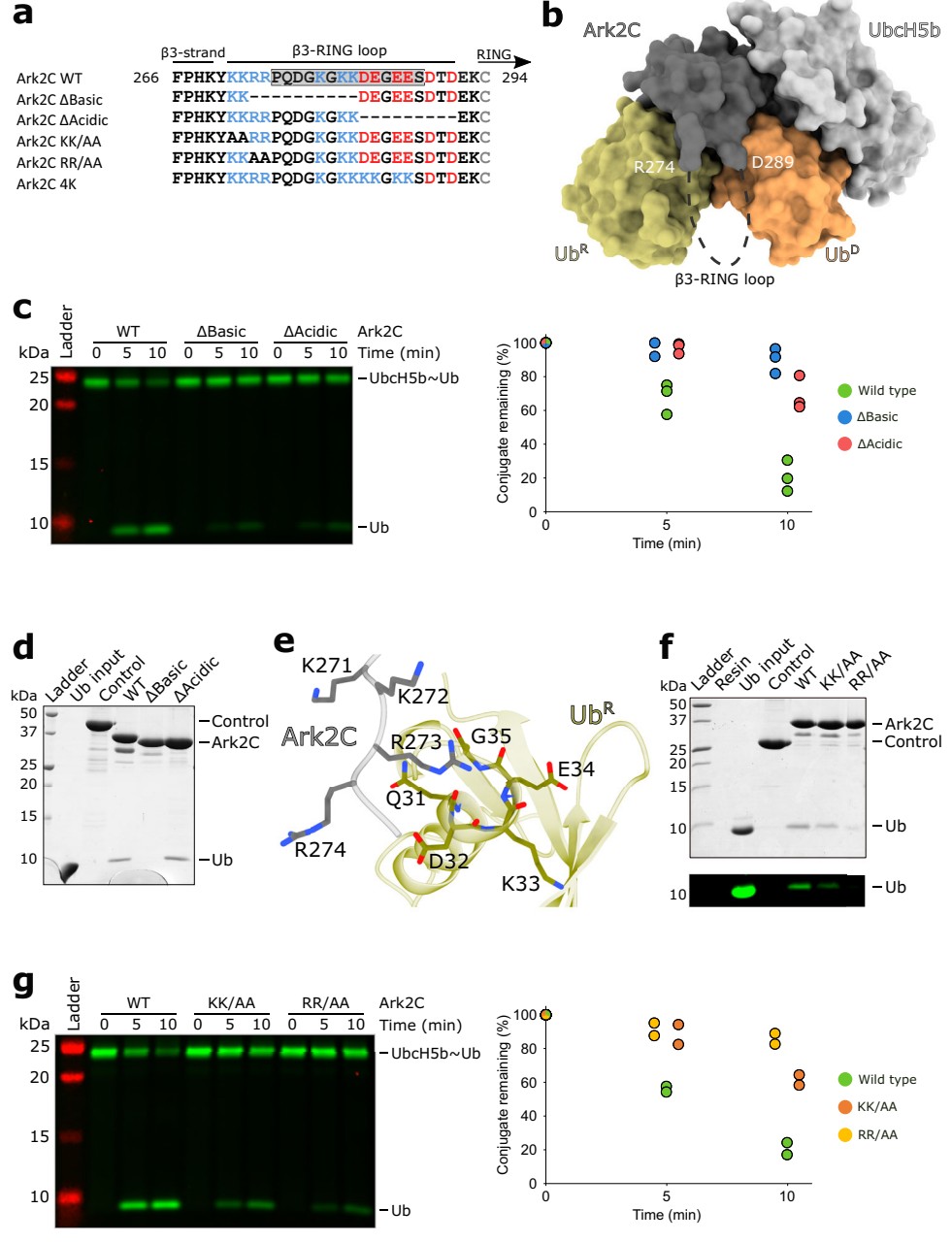

**Fig. 4 Analysis of the β3-RING loop and its involvement in coordinating Ub^R. a** Sequences of the β3-RING loop constructs and mutants used in this study. Residues in the grey box in the WT sequence were missing from the crystal structure. **b** Surface diagram of the activated complex with the missing residues in the β3-RING loop shown as a dashed line (black) and the position of the last modelled residues in the crystal structure indicated. **c** Discharge assays assessing the ability of the indicated Ark2C deletion mutant proteins to promote hydrolysis of the UbcH5b-Ub conjugate. Ubiquitin used to prepare the UbcH5b~Ub conjugate was labelled and the gel was imaged as described in Fig. 1. The assay was performed in triplicate and the quantified data is shown to the right. **d** GST-pulldown assay comparing the ability of the Ark2C-ΔBasic and -ΔAcidic deletion constructs to bind ubiquitin. **e** Close-up view of the contacts between the basic residues at the N-terminus of the β3-RING loop and Ub^R in the UbArk2C structure. **f** GST-pulldown assay to assess the contribution of basic residues to ubiquitin binding. **g** Discharge assay as in panel (**c**) evaluating the ability of the mutant proteins to promote ubiquitin release. The assay was done twice and the discharge of conjugate was quantified. Both **c** and **g** were performed using 0.25 μM E3 ligase and 5 mM ʟ-lysine. Source data are provided in the Source Data file.

both mutants had diminished activity in single turnover UbcH5b~Ub hydrolysis assays (Fig. 4g). These results suggest that the basic residues at the N-terminal end of the β3-RING loop extend the ubiquitin binding surface on Ark2C, which promotes Ub^R recruitment and ubiquitin transfer.

**Acidic residues in the β3-RING loop enhance ubiquitin transfer.** Initial analysis of the β3-RING loop deletion constructs

showed that the acidic residues are important for activity, but not required for recruitment of Ub^R (Fig. 4c, d). Analysis of the surface electrostatic potential (Fig. 5a and Supplementary Fig. 5e) and recent studies[34,35] suggested that the acidic residues might interact with basic residues (Lys11, Lys29 and Lys33) in Ub^D that are adjacent to the RING domain. To investigate this, the first Lys11 in the β1-β2 loop was mutated to Ala (Ub^K11A) and Asp (Ub^K11D) and the activity was compared. UbArk2C was used for

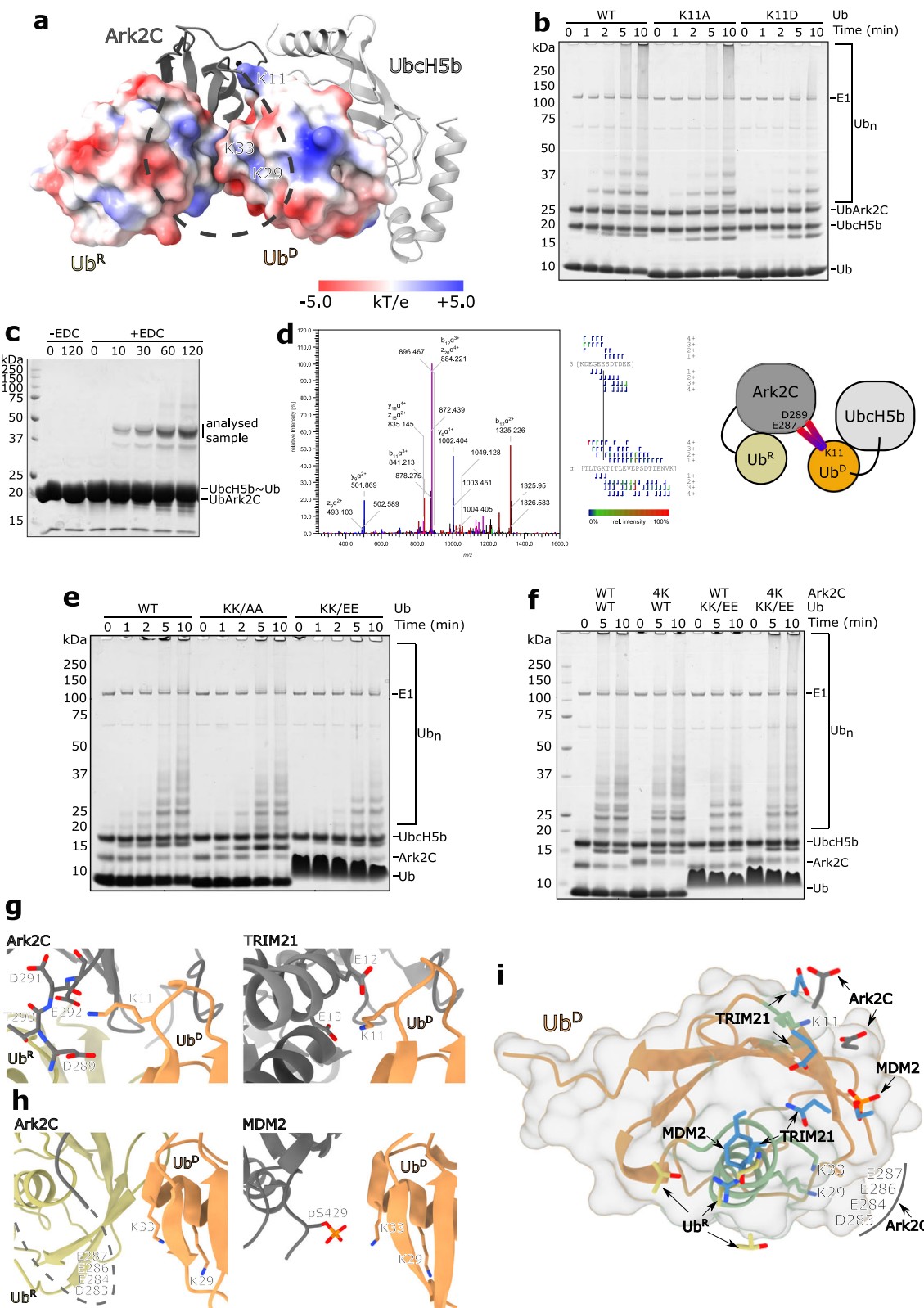

this analysis as we have previously established that Lys11 in Ub^R is required for recruitment of the E2~Ub conjugate and activity. In chain-building assays we observed a decrease in activity for both Lys11 ubiquitin mutants compared to when WT ubiquitin was used, with the reduction most obvious for Ub^{K11D} (Fig. 5b). Furthermore, when UbcH5b conjugates were prepared with the Ub^{K11A} and Ub^{K11D} variants a reduced level of discharge was

observed for Ub^{K11D} (Supplementary Fig. 5f). These assays suggest that Lys11 in Ub^D is important for activity with Ark2C. Indeed, when we prepared the UbArk2C–UbcH5b~Ub complex and assessed interactions using a crosslinker, EDC, that links acidic residues to basic residues that are in very close proximity as it has a zero-length spacer, we identified the highest-scoring crosslinking between Lys11 of ubiquitin and Glu287 of Ark2C

**Fig. 5 Acidic residues in the β3-RING loop make extensive contacts with Ub$^D$. a** UbcH5b and Ark2C are represented as grey cartoons with the basic β3-RING region shown as a black dashed line. Ub$^R$ and Ub$^D$ are shown as a surfaces coloured to highlight their electrostatic potential with contours at ±5.0 kT/e. The position of relevant residues is highlighted. **b** Multiturnover assays showing that mutation of Lys11 to Ala (K11A) and Asp (K11D) impedes activity. For these assays, UbArk2C was utilised to ensure the Ub$^R$ binding site was occupied by WT ubiquitin. **c** A 1:1 UbArk2C–UbcH5b~Ub complex (50 μM) was incubated with EDC crosslinker for 45 min and then resolved by SDS-PAGE. **d** Analysis of the excised band revealed a cross-link between Lys11 in ubiquitin and Glu287/Asp289 in the β3-RING loop. MS/MS spectrum of the EDC cross-linked peptide with relevant b and y fragment ions highlighted on the spectrum and the peptide sequence. The colours of the fragment marker indicate its relative intensity. **e** Multiturnover assays comparing the activity of WT Ark2C with WT ubiquitin and ubiquitin in which Lys29 and Lys33 were mutated to Ala (K29A/K33A) and Glu (K29E/K33E) as indicated. **f** Multiturnover assays comparing the activity of WT Ark2C and a mutant in which the acidic residues (Asp283, Glu284, Glu286 and Glu287) were mutated to lysine (referred to as 4 K). Both WT ubiquitin and a mutant in which Lys29 and Lys33 were mutated to Glu (KK/EE) were utilised. Gels were stained with Coomassie Blue and molecular weight standards are indicated. **g** Close-up view of the contacts between the TRIM21 RING and Lys11 in Ub$^D$ of the TRIM21–E2–Ub complex (PDB: 6S53), and the equivalent region of the UbArk2C–UbcH5b~Ub structure showing interactions between the C-terminal residues in the β3-RING loop and Lys11 in Ub$^D$. **h** Structure of MDM2 in complex with an E2-Ub conjugate (PDB: 6SQS) showing contacts between a phosphorylated residue (Ser429) and Lys33 in Ub$^D$ (left) and the equivalent region from the UbArk2C–UbcH5b~Ub structure showing the position of Lys33 in Ub$^D$. The β3-RING loop is shown as a dotted line and the acidic residues in the centre of the loop are indicated. **i** Ub$^D$ is shown as a ribbon with a surface and the key contacts that stabilised the activated conformation in MDM2, TRIM21 and Ark2C are shown. The figure was prepared by superimposing Ub$^D$ from the activated complexes with MDM2 (PDB: 6SQS), TRIM21 (PDB: 6S53) and Ark2C. Key contact residues from the E3s are shown as blue sticks. Contacts mediated by Ark2C bound Ub$^R$ from this study are shown as grey and yellow sticks, respectively.

(Fig. 5c, d), followed by the second-highest scoring crosslink between Lys11 and Asp289. The reduced spacer arm of the cross-linker and structural data together indicates that this interaction is between Ark2C and Ub$^D$.

Next, we assessed the importance of Lys29 and Lys33 for chain assembly by mutating them to alanine (Ub$^{KK/AA}$) and glutamic acid (Ub$^{KK/EE}$). When these ubiquitin variants were used in chain-building assays with Ark2C, a marked reduction in activity was observed for Ub$^{KK/EE}$, while the Ub$^{KK/AA}$ mutant had an intermediate level of activity (Fig. 5e). In the structure, residues 275–288 in the β3-RING loop were not resolved, but it seemed possible that the four acidic residues (at positions Asp283, Glu284, Glu286 and Glu287) could interact with Lys33 and Lys29 on the helix of Ub$^D$ (Fig. 5a). To investigate whether a direct interaction might occur we used a charge swap experiment, where switching the charges on one protein is expected to reduce activity, but when two mutated proteins with the complementary changes are mixed, activity is recovered. We replaced the four acidic residues with lysine (Ark2C$^{4K}$) and analysed its activity with WT ubiquitin and Ub$^{KK/EE}$. In chain building assays a modest reduction in activity was observed when Ark2C$^{4K}$ was paired with WT ubiquitin. However, when Ub$^{KK/EE}$ was used together with Ark2C$^{4K}$, activity increased and was more comparable to that observed when the WT proteins were paired (Fig. 5f). This result suggests that Lys29 and Lys33 on the α-helix of ubiquitin interact with the acidic residues in the centre of the β3-RING loop of Ark 2C (Fig. 5f) and that this interaction enhances ubiquitin transfer by Ark2C.

The interactions between basic residues in Ub$^D$ and acidic residues in Ark2C resemble those observed between Ub$^D$ and both TRIM21 and MDM2[34,35], where the functionally equivalent acidic groups directly contact the basic sidechains in Ub$^D$ (Fig. 5g–i). However, in Ark2C it seems that there is greater flexibility in this region with multiple contacts supporting activity. For example, the sidechain of Lys11 crosslinked to both Asp289 and Glu292 indicating that they are in close proximity, even though Glu292 was disordered in the crystal structure. Furthermore, the four central acidic residues in the loop were not resolved in the structure, but the biochemical data support interaction with Lys29/Lys33 in Ub$^D$. Together, these results suggest that basic residues in Ub$^D$ interact with several acidic residues in the β3-RING loop of Ark2C to promote ubiquitin transfer (Fig. 5g, h), most probably because they help to stabilise the activated conformation of the bound E2~Ub conjugate.

## Discussion

Transfer of ubiquitin from an E2 to a substrate represents the crux of the ubiquitin cascade and has a pivotal role in the control of many cellular signalling events. Not surprisingly this step is tightly regulated, with activation of the E2~Ub conjugate required for ubiquitin transfer[28]. For RING–E3s, the interaction of the RING domain with the conserved UBC domain of the E2 is the primary activating allosteric step and is sufficient to promote nucleophilic attack and formation of an isopeptide bond[3]. However, in recent years it has become apparent that elements beyond the core RING domain often enhance E3 ligase activity[3]. Here, we report a detailed analysis of ubiquitin transfer by Ark2C, an E3 that possesses a bone fide ubiquitin-binding RING domain. The structure of the activated complex reveals the key regulatory role played by Ub$^R$ in the stabilisation of the closed and primed conformation of the E2~Ub conjugate (Fig. 1c, d), and it suggests a Ub$^A$-binding site that supports the assembly of Lys48-linked chains by UbcH5b (Fig. 2a). The importance of the charged loop in helping to recruit Ub$^R$ and contacting Ub$^D$ to facilitate activity is also revealed.

Previously we discovered that ubiquitin transfer by the monomeric E3s, Arkadia and Ark2C, was enhanced by binding of ubiquitin to the RING domain and we reported structures of the RING domain bound to Ub$^R$ and a E2~Ub conjugate, but neither structure was primed for ubiquitin transfer[23]. The activated structure of the Ark2C–E2~Ub complex reported here shows that adjustments in the position of Ub$^R$ (Supplementary Fig. 3d), relative to the non-primed structures, allows for extensive contacts between Ub$^R$ and the α-helix in Ub$^D$ (Fig. 1d). This explains why Ark2C activity is reduced when the binding of Ub$^R$ is disrupted. Furthermore, the contacts between Ub$^R$ and the α-helix in Ub$^D$ overlap with contacts provided by the dimeric partners in many RINGs (Fig. 5i). For example, Lys11 of Ub$^R$ and the key aromatic residue in MDMX[36], as well as other dimeric RING–E3s such as RNF4[10], and IAPs[11], occupy similar sites. As a result, the Ub$^R$ molecule mimics the dimeric RING partner by contacting the C-terminal end of the α-helix in Ub$^D$ (Fig. 5i).

Comparison of available structures of monomeric RING domains from RNF13, RNF38[32], RNF12[37], PJA1 and ZNRF1[38] with Ark2C highlights a conserved RING structure, that includes an additional N-terminal β-strand (Supplementary Fig. 6a, b). The loop that connects this β-strand to the RING domain, denoted as the β3-RING loop, is variable in length and sequence, but it seems possible that it may have a conserved function. Like Ark2C, the C-terminal end of the β3-RING loop in these

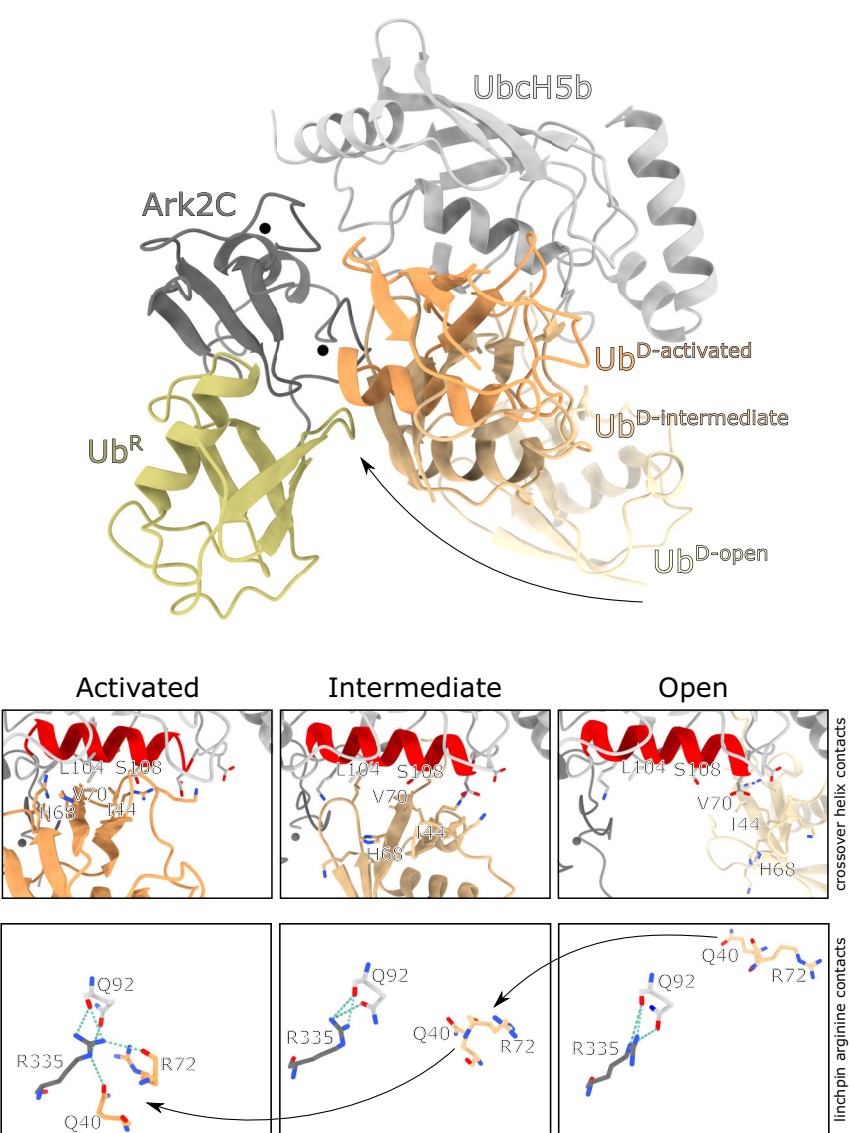

**Fig. 6 Multiple states of the bound E2-Ub conjugate.** The structures of Ark2C in complex with UbcH5b~Ub conjugate in an open (PDB: 5D0K), closed but not primed (PDB: 5D0M) and closed and primed conformation was overlayed. The primed complex structure is shown together with Ub$^D$ for all three structures as indicated. Ub$^R$ from the open and primed structures is also included to show the displacement that occurs upon binding. Below the contacts between Ub$^D$ and the crossover helix of the E2, as well as the linchpin arginine, are highlighted. The buried surface area for Ub$^D$ in the three structures is 170, 640 and 970 Å$^2$ for the Open, Intermediate and Activated states.

monomeric RINGs includes at least one acidic or hydrophilic residue that has the potential to contact Lys11 in Ub$^D$ (Supplementary Fig. 6a, b). It is also interesting to note that phosphorylation of a Tyr residue in an α-helix that is further N-terminal to the RING domain of CBL-b makes similar contacts with Lys11 in Ub$^D$ (Supplementary Fig. 6c)[8]. In a similar manner, phosphorylation of hydrophilic residues in this region of the monomeric RINGs may also regulate activity, and these residues could interact with Lys33 in Ub$^D$ to promote ubiquitin transfer as observed for MDM2[35]. Increasing evidence suggests that contacts with lysine residues in Ub$^D$ may be a common feature of RING E3s, even if the location of the interacting element is not conserved.

The structures now available for Ark2C in complex with the UbcH5b~Ub conjugate capture several of the possible conformations bound conjugates can adopt. Notably, they reflect the dynamic nature of the conjugate[39,40], with open and activated states available, as well as an intermediate state that is closed, but not primed (Fig. 6). This is consistent with prior studies that

suggested when the UbcH5b~Ub conjugate is bound to an E3, it is not necessarily restricted to a defined conformation[41]. Analysis of the Ark2C structures suggests that some contacts are common to all states, but other contacts are required to activate UbcH5b~Ub for nucleophilic attack (Fig. 6). For example, while the linchpin Arg of Ark2C (Arg335) contacts Gln92 of UbcH5b in all three structures, only in the activated complex does Arg335 make the essential contacts with Arg72 and Gln40 in Ub$^D$ to prime ubiquitin for transfer. Furthermore, while Ub$^D$ interacts with the crossover helix of UbcH5b in both the intermediate (buried surface area: ~640 Å$^2$) and activated structures (buried surface area: ~975 Å$^2$), the contacts are less intimate when not primed. The order in which contacts with the conjugate occur remains uncertain, but our studies with Ark2C suggest that Ub$^R$ and the β3-RING loop are required to stabilise the activated conformation so that maximal activity is achieved.

The formation of polyubiquitin chains is essential for the functional diversity of ubiquitin signals. In the case of RING E3 ligases, specification of chain linkage is attributed to the E2

**Table 1 Data collection and refinement statistics.**

| Data collection and refinement statistics | UbArk2C (PDB: 7R70) | UbArk2C-UbcH5b-Ub (PDB: 7R71) |
|---|---|---|
| *Data collection* | | |
| Space group | P 6$_2$ 2 2 | P 2$_1$ 2$_1$ 2 |
| *Cell dimensions* | | |
| a, b, c (Å) | 154.47, 154.47, 81.71 | 53.33, 75.67, 99.38 |
| α, β, γ (°) | 90, 90, 120 | 90, 90, 90 |
| Resolution (Å) | 44.59-2.499 (2.589-2.499) | 43.59-2.8 (2.9-2.8) |
| R$_{merge}$ | 0.1905 (3.075) | 0.1541 (1.239) |
| I/σI | 23.25 (1.61) | 13.74 (2.31) |
| Completeness (%) | 99.79 (98.74) | 99.70 (99.70) |
| Redundancy | 42.9 (42.5) | 13.2 (14.0) |
| CC 1/2 | 1 (0.685) | 0.999 (0.918) |
| Wilson B-factor | 54.9 | 56.6 |
| *Refinement* | | |
| Resolution (Å) | 2.5 | 2.8 |
| No. of reflections | 20,392 | 10,384 |
| R$_{work}$/R$_{free}$ | 0.198/0.242 | 0.224/0.269 |
| *No. of atoms* | | |
| Protein | 2386 | 2978 |
| Ligand/ion | 28 | 2 |
| Water | 22 | 0 |
| *B-factors* | | |
| Protein | 72.26 | 75.60 |
| Ligand/ion | 95.73 | 54.26 |
| Water | 57.03 | – |
| *R.m.s. deviations* | | |
| Bond lengths (Å) | 0.007 | 0.004 |
| Bond angles (°) | 1.20 | 0.98 |

Statistics for the highest-resolution shell are shown in parentheses.

molecules[42,43], which bind Ub$^A$ and orient specific lysine residues towards the active site. To date, only acceptor binding sites for the assembly of Lys11-linked chains by UbcH5a and Lys63-linked chains by Ubc13 have been observed in crystal structures[25,44]. In the UbArk2C–UbcH5b~Ub complex structure crystal packing results in interaction of ubiquitin from one complex near the active site of UbcH5b in a neighbouring complex. Given the proximity of Lys48 to the active site in the structure and the ability of UbcH5b to efficiently assemble Lys48-linked ubiquitin chains[25,26], it seems that ubiquitin positioned in this way would support the assembly of Lys48 chains. Rearrangements would be required as Lys48 is not pointing at the active site, but Lys48 resides on the highly mobile β3–α2 loop in ubiquitin and could therefore access the active site (Fig. 2c)[27]. Other E2-Ub$^A$ interactions have also been proposed to support Lys48 assembly by Ube2K and Ube2R1[45,46], although in all cases interaction of the acceptor with α3-helix of the E2 seems important. This suggests, that Ub$^A$ can approach the active site in different ways to achieve the formation of a Lys48 link.

From this study, it is clear that, when paired with UbcH5b, the activity of Ark2C is enhanced by the interaction of ubiquitin at both the Ub$^R$ and Ub$^B$ binding sites (Fig. 3a). While the role of these sites within a cellular setting is not well understood, it seems possible that the dependence of activity on ubiquitin binding may provide a very elegant way to fine-tune the activity of Ark2C and Arkadia. For instance, substrates that have been mono-ubiquitylated or multi-monoubiquitylated would be expected to bind more tightly to the E3 and to activate the E3, resulting in their extensive ubiquitylation. These interactions could also help tether the substrate to the RING domain as chain extension

occurs. In this way, the reliance of Ark2C and Arkadia activity on ubiquitin binding may provide a sophisticated mechanism to ensure processive modification of substrates and rapid amplification of the ubiquitin cascade.

## Methods

**Constructs, protein expression and purification.** All recombinant proteins were expressed in *Escherichia coli*, strain BL21 (DE3) using 0.2 mM IPTG (APA4773.0025, AppliChem) for induction. For Ark2C (RNF165), UbArk2C (Fig. 1b), and UbcH5b (Ube2D2) the coding region for the human sequences were cloned into pGEX-6P-3 (GE Healthcare) resulting in the expression of GST-fusion proteins. GST-fused proteins were purified using Glutathione Sepharose™ resin (GE Healthcare) in phosphate-buffered saline (PBS: 137 mM NaCl, 2.7 mM KCl, 10 mM Na$_2$HPO$_4$, and 1.8 mM KH$_2$PO$_4$) pH 7.4 as described previously[23]. In brief, the GST tag was cleaved using GST-fused rhinovirus (HRV) type-14 3C protease which left five additional amino acids (GPLGS) at the N-terminus of each fusion protein. The cleaved proteins were further purified by size-exclusion chromatography (SEC) using a Superdex™ 75 column (GE Healthcare) pre-equilibrated in PBS.

Ubiquitin was cloned into pET-3a, expressed as an untagged protein and then purified using a protocol derived from Sato et al.[45]. Briefly, ubiquitin was purified on a HiTrap™ SP HP (GE Healthcare) equilibrated in 50 mM ammonium acetate, pH 4.5 with 1 mM EDTA and eluted with a linear 0–1 M NaCl gradient. Fractions containing ubiquitin were pooled and further purified on a Superdex™ 75 column (GE Healthcare) equilibrated in 20 mM Tris-HCl pH 7.5, and 150 mM NaCl. The ubiquitin variants used for Cy3-labelling, which have an additional Met and Cys residue at the N-terminus, were purified using the same protocol. The Cys residue was labelled according to the manufacturer's protocol for Cy3 Maleimide Mono-Reactive Dye (PA23031, GE Healthcare). All mutations and deletions were made using a protocol based on that described by Liu and Naismith[46].

**Preparation and analysis of UbcH5b-Ub conjugates.** To prepare thioester E2~Ub conjugates for assays, WT UbcH5b (100-200 μM) was incubated with E1 (1 μM) and ubiquitin (300-600 μM) at 37 °C for 40 minutes in 20 mM HEPES pH7.5, 50 mM NaCl, 2 mM MgCl$_2$, 2 mM TCEP (C4706, Sigma-Aldrich) and 10 mM ATP (A7699, Sigma-Aldrich). The reaction products were subsequently separated using a Superdex™ 75 10/300 GL column equilibrated in 20 mM HEPES pH 7.5, 150 mM NaCl. The eluted conjugate was analysed by SDS-PAGE, and the fractions corresponding to the E2~Ub conjugate were pooled and concentrated to ~100 μM before snap-freezing.

Isopeptide linked UbcH5b~Ub conjugate for crystallography and crosslinking purposes was prepared using wild-type ubiquitin and UbcH5b, which included mutations to allow isopeptide conjugate formation (C85K), prevent backside binding (S22R) and increase E2 stability (C21S, C107S, C111S). The reaction contained E1 (1 μM), E2 (100–200 μM), ubiquitin (300–600 μM) and ATP (4 mM) (A7699, Sigma-Aldrich) in cycling buffer [25 mM Tris pH 9.5, 2.5 mM MgCl$_2$, 5 mM phosphocreatine (P7936, Sigma-Aldrich) and 0.6 U/mL creatine phosphokinase (C3755, Sigma-Aldrich)]. Typically a 5 mL reaction was incubated at 37 °C for 18-24 hours then separated on a HiLoad™ 26/600 Superdex™ 75 prep column, pre-equilibrated in 20 mM Tris-HCl pH 7.5, 200 mM NaCl[47].

**Crystallisation.** For crystallography purposes, UbArk2C was purified as described above with an additional ion-exchange step to improve purity. Fractions collected after SEC was loaded onto a HiTrap™ Q HP (GE Healthcare) equilibrated in 20 mM Tris, pH 7.5, 50 mM NaCl and eluted with a linear 50–1000 mM NaCl gradient.

For UbArk2C, single crystals were obtained by mixing purified UbArk2C at 30 mg mL$^{-1}$ with 1.6 M Na$_3$Citrate (Molecular Dimensions) at a protein:well solution ratio of 1:2 (200 nL:400 nL). All crystals were grown by vapour diffusion as sitting drops in Swissci 3 Lens Crystallisation Plate (HR3-125, Hampton Research) at 16 °C. This yielded a number of single crystals that were further improved by utilising a ratio of 2:1 (400 nL:200 nL). Crystals were cryoprotected with 30% glycerol in the original condition before flash-freezing. Diffraction data were collected using 13 keV photons at 100 K, allowing 360° rotation with 1° oscillation per image at MX1 beamline of the Australian Synchrotron[48].

The co-purified complex of UbArk2C-UbcH5b~Ub, concentrated to ~5 mg/mL, was the subject of initial crystal trials set up at protein:well solution ratio of 1:1 (200:200 nL). Clusters of needles crystallised in 0.1 M MES pH 6.0, 20% w/v PEG 6000, and 0.01 M ZnCl$_2$. These were further improved by the addition of 3% w/v Xylitol from the additive screen (HR2-138, Hampton Research). The resulting clusters were crushed using polytetrafluoroethylene Seed Bead™ (HR2-320, Hampton Research) for Microseed Matrix Screening. Crystal seeds were mixed with the purified complex, and JCSG-plus™ and PACT premier™ screens in ratio 1:5:3 of seed:protein:well solution (50:250:150 nL). Microseed matrix screening yielded three-dimensional crystals in 0.1 M MMT buffer pH 6.0, 25% w/v PEG 1500. Data collection were performed at the MX2 beamline of the Australian Synchrotron[49], resulting in a 2.8 Å dataset. Diffraction data were collected at 100 K, using energy of 13 keV, and allowing 360° rotation with 0.1° oscillation per image.

**Structure determination**. For the UbArk2C structure, the data was processed to 2.5 Å using XDS[50], with subsequent merging and scaling performed using Aimless[51] (CCP4 suite). The structure of UbArk2C was solved by molecular replacement using Phaser MR[52] (CCP4 suite) and the structure of Ark2C bound to Ub[R] (PDB: 5D0K) as a search model. UbArk2C crystallised in space group P 6$_2$ 2 2 with two copies in the asymmetric unit. The UbArk2C structure was iteratively improved by manual adjustments using Coot, followed by refinement in Phenix[53] to a final $R_{work}/R_{free}$ = 0.198/0.242 (Table 1).

The UbArk2C-UbcH5b~Ub complex structure was determined to 2.8 Å using a workflow similar to that described for UbArk2C. For molecular replacement, the UbArk2C structure and the UbcH5b~Ub conjugate in the activated conformation from PDB: 4V3L were used. The activated complex of UbArk2C–UbcH5b~Ub crystallised in space group P 2$_1$ 2$_1$ 2 with a single copy in the asymmetric unit. The complex was refined using Phenix[53] to $R_{work}/R_{free}$ = 0.224/0.269 (Table 1). UCSF ChimeraX was used to prepare structural figures[54]. Crystal contacts were analysed using PISA[55].

**Ubiquitylation assays**. Multiturnover activity assays were carried out at 37 °C for the indicated times in 20 mM Tris-HCl pH 7.5, 50 mM NaCl, 2 mM MgCl$_2$, 2 mM DTT (1114GR005, BioFroxx), and 10 mM ATP (A7699, Sigma-Aldrich), containing E1 (0.1 μM), E2 (10 μM), E3 (5 μM), and ubiquitin (50–100 μM) (final concentration). The reactions were incubated at 37 °C and stopped by the addition of a reducing SDS sample buffer. For some assays, Cy3-labelled ubiquitin was used in 1:1 ratio with the unlabelled form[9]. Following separation by SDS-PAGE reactions containing Cy3-labelled ubiquitin were imaged with an Odyssey® Fc (LI-COR Biosciences) using a 600 nm filter prior to Coomassie® Brilliant Blue R-250 (APA1092.0025, AppliChem) staining.

For single-turnover ubiquitin discharge assays, 15 μM of E2~Ub conjugate variants were incubated with 0.125–5 mM L-lysine (L5501, Sigma-Aldrich) and 0.25–1 μM of E3 ligase. All assays were incubated at 25 °C and then stopped by mixing with non-reducing 2× SDS sample buffer, which ensured that the unhydrolyzed thioester E2~Ub conjugate remained intact. Samples were resolved by SDS-PAGE and the reactions containing Cy3-labelled E2~Ub conjugate were imaged by an Odyssey® Fc (LI-COR Biosciences) using 600 nm filter prior to Coomassie® Brilliant Blue R-250 (APA1092.0025, AppliChem) staining.

**Binding studies**. GST pull-down assays were performed by mixing GST-Ark2C proteins immobilised on Glutathione Sepharose™ resin (GE Healthcare) with ubiquitin. Samples were made up to 200 μL in a buffer containing PBS pH 7.4, 0.2% Tween® 20 (P7949, Sigma-Aldrich), and 2 mM DTT (1114GR005, BioFroxx), and incubated on a tube rotator at 4 °C for 30 min. Samples were then washed four times with PBS before being mixed with reducing 2× SDS sample buffer and resolved by SDS-PAGE. The pulldowns containing Cy3-labelled ubiquitin were imaged using an Odyssey® Fc (LI-COR Biosciences) with a 600 nm filter prior to Coomassie staining.

Analytical SEC was performed using a 10/300 Superdex 75 increase (GE Healthcare) equilibrated with 20 mM Tris-HCl pH 7.5, 200 mM NaCl buffer. For each run, 100 μL of 200 μM of purified Ark2C variant, ubiquitin, or a mixture of the two proteins was resolved. For mixtures, samples were incubated overnight prior to separation. The recovered fractions were analysed by SDS-PAGE.

**Crosslinking and mass spectrometry**. The amine-crosslinking reaction was performed by mixing UbArk2C and UbcH5b~Ub at ~50 μM each with BS³-d$_0$ crosslinker (21590, ThermoFisher) at a ratio of 1:1:40. The reaction was incubated at room temperature for 45 min, then quenched using 1 M Tris-HCl pH 7.5 for 15 min prior to mixing with reducing 2× SDS sample buffer and separation by SDS-PAGE. A similar approach was used for the carboxyl-/amine-crosslinking, with the EDC zero-length crosslinker (22980, ThermoFisher) mixed with the complex at a ratio 1:1:100 in 50 mM MES pH 6.5 and allowed to react for up to 2 h. Reactions were resolved by SDS-PAGE. Selected bands were excised and subjected to in-gel digestion with trypsin. Following digestion peptide samples were injected onto an Ultimate 3000 nano-flow uHPLC-System (Dionex Co, USA) coupled with the linear trap quadrupole (LTQ)-Orbitrap XL hybrid mass spectrometer (Thermo Scientific, USA). Peptides were separated on an in-house packed 20 cm emitter-tip column [75 μm ID (Phoenix S&T, USA) packed with 2.6 μm Aeris C-18 beads (Phenomenex, USA)] and eluted from the column using a 5–99% v/v acetonitrile gradient. The mass spectrometer was operated in full MS in between 100–2000 m/z in the Orbitrap mass analyser at a resolution of 60,000. The strongest 11 signals between 400 and 2000 m/z were selected for collision-induced dissociation MS in the LTQ ion trap.

For peptide crosslink identification the MS/MS data were searched against an in-house sequence database consisting of UbcH5b~Ub and Ub-Ark2C sequences using StavroX software[56]. The mass of the BS3 crosslinker (138.06 Da) was enabled for lysine. Carbamidomethyl cysteine, oxidised methionine and deamidation of asparagine and glutamine were included as variable modifications. The precursor mass tolerance threshold and the maximum fragment mass error was set at 5 ppm and 0.5 Da, respectively. The minimum false discovery rate cut-off and score cut-off were set at 5% and 50%, respectively. The mass range of crosslinked peptides was set at a range of 200–6000 Da. Following sorting, the MS/MS spectrum of the resulting crosslinked peptides of interest was visually checked for the presence of the relevant MS/MS peaks.

The presence of b and y ion peaks were checked to confirm the peptide sequence and the mass of the corresponding crosslinked peptide.

**Statistics and reproducibility**. All experiments were repeated two or more independent times with similar results. Blot and gels were processed and analysed with Image Lite v5.2.5. Figures were assembled using Inkscape v1.1.

**Reporting summary**. Further information on research design is available in the Nature Research Reporting Summary linked to this article.

## Data availability

Atomic coordinates and structure factors have been deposited in the Protein Data Bank with accession codes of 7R70 (UbArk2C fusion) and 7R71 (UbArk2C–UbcH5b~Ub complex). This study uses publicly available data from the PDB under accession codes: 2KN5, 6SQS, 6S53, 5D0K, 5D0M, 5VO0, 6HPR, 5H7S, 5MNJ, 5EYA, 4AP4, 3ZNI, 4V3L, 4AUQ, 6W9D, 5ZBU, 2L0B and 5YWR. Source data are provided with this paper. All raw data (e.g. uncropped, unannotated gels and western blots) corresponding to individual figure panels are provided in the Source Data File. All unique biological materials used are available upon reasonable request. Source data are provided with this paper.

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

## Acknowledgements

We thank Prasanth Padala for assistance with the crosslinking experiments, and Abhishek Kumar, Karen Knapp and Torsten Kleffmann from the Centre for Protein Research (University of Otago) for mass spectrometry support. AP acknowledges the support of a University of Otago Doctoral Scholarship. Financial support from the Marsden Fund (NZ) of New Zealand to CLD is also gratefully acknowledged.

This research was undertaken using the MX1 and MX2 beamlines at the Australian Synchrotron, part of ANSTO, and made use of the Australian Cancer Research Foundation (ACRF) detector. We thank the New Zealand Synchrotron Group for facilitating access to the Australian Synchrotron.

## Author contributions

A.P. performed all structural experiments and most biochemical analyses. A.J.M. contributed to the structural studies. C.R. completed some assays. A.P., A.J.M., C.R., P.M. and C.L.D. contributed to experiment design, analysed data and provided feedback on the manuscript. C.L.D. obtained funding supervised the study and wrote the paper together with A.P.

## Competing interests

The authors declare no competing interests.
