## [Peer Review File · Nature Communications]

Ubiquitin and a charged loop regulate the ubiquitin E3 ligase activity of Ark2CREVIEWER COMMENTS

Reviewer #1 (Remarks to the Author):

This is a well-written, clearly communicated study, reporting the finding of elements within the Ark2C RING (and by conservation, Arkadia) that regulate ubiquitin transfer through interactions with catalytic and non-catalytic ubiquitin molecules. The Day lab has a well-regarded track record in understanding the finer details of ubiquitin molecules regulating ubiquitin transfer processes, and this work builds upon and extends earlier findings.

The main results are the determination of a complex of Ark2C RING in complex with E2-Ub, and also bound to another ubiquitin molecule by virtue of a fusion to the N-term of the Ark2C construct. The quality of the structural biology is high, and the findings are supported by careful biochemistry that validate the interactions between the proteins, and the role those interactions play in the transfer process.

In this reviewer's opinion, the data are sound, the conclusions are supported by the experimental results presented, it is not 'oversold' as being the fundamental way all E3 ligases work, rather it is carefully discussed in the context of the functional roles the ligases play. I found the figures easy to follow, and have no major concerns. I have one curiosity question which the authors may be able to address, which is given the propensity for Arkadia to bind SUMO and polySUMO chains, do they think that a fusion of SUMO instead of UbR might achieve the same stabilisation of the activated E2-Ub they observe in their structure?

A few minor points -

p7. line 167 - end of sentence may need a specific reference to support the Lys11 statement.

p7. line 171 - should be burying 375 Å² not just Angstrom (missing a squared symbol)

p7. line 191 - and in materials and methods - when describing the S22R version of UbcH5 that cannot build polyubiquitin chains, there should be a reference to the original paper describing this - Hibbert and Sixma, J Biol Chem 2012 Nov 9;287(46):39216-23. doi: 10.1074/jbc.M112.389890.

Intrinsic flexibility of ubiquitin on proliferating cell nuclear antigen (PCNA) in translesion synthesis

PMID: 22989887

Reviewer #2 (Remarks to the Author):

Paluda et al. have solved the crystal structure of a ubiquitin Ark2C linear fusion in complex with an E2 ubiquitin conjugate. This follows a paper by Wright et al. in NSMB in 2016, where a model of such a complex was proposed. In this structure the E2 ubiquitin conjugate is captured in the closed conformation indicating an active complex. As Ark2C is functional as a monomeric RING E3 ligase, the closed conformation of the E2 ubiquitin conjugate is stabilized by the linearly fused ubiquitin to Ark2C, which binds to the ubiquitin molecule conjugated to the E2. Biochemical analysis of mutations at this interface confirms the importance of this interaction for catalysis. The crosslinking mass spectrometry experiment reveals an interaction between the β 3-RING loop of Ark2C and ubiquitin. The authors also propose a model for K48 linked ubiquitin chain formation based on crystal packing.

Structural results confirm non-covalent ubiquitin as a modulator of RING E3 ligase activity and mass spec crosslinking exposed a previously unknown role for the β 3-RING loop. In the title of the manuscript, there are two elements being investigated: the effect of non-covalent ubiquitin and the effect of the β 3-RING loop of Ark2C on catalysis. However, the biochemical analysis presented in the paper is not sufficient to validate the importance of specific interactions predicted from the structure and does not clearly dissect out the separate roles of the regulatory ubiquitin and the β 3-RING loop on Ark2C catalysis.

Specific points:

1. Figure 1b and c: It is not clear in these figures and not mentioned in the main text on page 5 which residues of the Ubiquitin Ark2C linear fusion are resolved and not resolved in the crystal structure. There are multiple termini in Ark2C in Figure 1c, however, there are no labels on the figure to help the reader understand what they correspond to. The authors need to label the figure and specify which residues are resolved.
2. Figure 1e: Ark2C alone (without ubiquitin linearly fused to it) is required as a control to show the effect of the absence of a non-covalent ubiquitin molecule. Considering Ark2C retains significant activity compared to the ubiquitin Ark2C linear fusion (Supplementary Figure 1c), it is therefore important that Ark2C alone is included in Figure 1e, to accurately determine the role of non-covalent ubiquitin in the mechanism of Ark2C.
3. Figure 1f: Again, Ark2C alone is required to show the effect of mutations in the E2 ubiquitin conjugate on Ark2C catalysis in the absence of non covalent ubiquitin.

4. Supplementary Figure 1c: Both Ark2C and ubiquitin Ark2C are tested in this figure and both appear to show similar catalytic rates. However, in the main text (page 5, line 116) the authors claim that ubiquitin transfer is affected. The difference appears to be in the length of ubiquitin chains formed, with ubiquitin Ark2C favouring longer ubiquitin chains. It is difficult to determine a difference between the amount of free ubiquitin consumed in the assay as ubiquitin is overloaded and not quantified. Quantitation is required to determine the effect on ubiquitin transfer. In addition, the band for Ubiquitin Ark2C is stronger than Ark2C alone. This may be due to differences in staining, however, it is important to ensure E3 ligase concentration is the same when comparing their activity.

5. Figure 2a: It is not clear from the figure whether K6 or K48 is closer to the E2 active site. Distance measurements are required in this figure.

6. Figure 2b: It is not clear in this assay that ubiquitin chain assembly was significantly reduced for the R125D mutant, as mentioned in the main text on page 7, line 174. Similar amounts of free ubiquitin are consumed and there is also an accumulation of even longer ubiquitin chains at the top of the gel for the R125D mutant. Quantitation is required to determine the effect on ubiquitin chain formation. In addition to K6 and K48 linked ubiquitin chains, K63 and K11 linked ubiquitin chains can be assembled by UbcH5b. To claim that this is a model for K48 linked ubiquitin chain formation, mass spec analysis of the mixture of chains needs to be performed on both WT and the R125D mixture to determine the type of ubiquitin chains formed and whether the R125D mutation alters the specificity.

7. Figure 3a: The S22R mutation, which perturbs backside binding of ubiquitin to UbcH5a, is more deleterious to longer ubiquitin chains formation than the M313A mutation, which perturbs UbR binding to Ark2C. Quantitation is required to determine the difference between WT and M313A and this finding should be reported and discussed in the main text.

8. Figure 4 a and b: As mentioned for Figure 1b and c, the authors need to specify which residues of the Ubiquitin Ark2C linear fusion are resolved and which residues are not resolved in the main text (page 8, line 207).

9. Supplementary Figure 5a: In the crosslinking mass spec experiment, the authors cannot distinguish between which ubiquitin molecule (UbR or UbD) the β 3-RING loop of Ark2C is crosslinking to. Comparing Ub-Arc2C and Ark2C in the cross-linking experiment would be useful in this respect, but the issue could be resolved by using, heavy isotope labeled UbD to set up the cross-linking analysis. This is an important point as differentiating between ubiquitin molecules is critical to the mechanism proposed.

10. Figure 4c: Ark2C alone is used in this discharge assay and given that Figure 1e shows the UbD and UbR interaction is very important it is surprising that discharge activity is observed in WT Ark2C alone. The Ubiquitin Ark2C linear fusion is required for comparison in Figure 4c and quantitation of the results is required as per Figure 1e.

11. Supplementary Figure 5c: The authors claim that chain building is impeded by the delta basic and delta acidic forms of Ark2C in this figure, however, long ubiquitin chains are still formed by both forms of Ark2C. Similar amounts of free ubiquitin are consumed in the assay also. Quantitation of this assay is required to determine if ubiquitin transfer is impaired as well as the formation of different lengths of ubiquitin chains.

12. Figure 5b and c: Removing the lysine side chain of K11, K29 and K33 in ubiquitin by mutating to alanine does not show any effect on activity in these figures. Considering only mutation to the opposite charge, either D or E, effects activity, this indicates that these residues are not contributing to specific interactions and mutating to opposite charge is resulting in repulsion. In addition, the fact that Ub KK/EE runs differently on the gel makes visual interpretation of the gel particularly difficult. As a result, the biochemical analyses to support the hypothesis that the acidic residues of the β 3-RING loop of Ark2C interact with UbD are weak.

In summary, the authors have determined an interesting structure which reveals the role of a bound regulatory ubiquitin and suggests a role for the loop between b3 and the RING that is not resolved in the structure. Further biochemical and mass spec experiments will be required to validate the role of the regulatory ubiquitin and determine the key interactions made by the b3-RING loop. Apart from the assays shown in Fig 1e, f the biochemical assays have not been subjected to quantitative analysis. This is important as the authors are making claims that particular residues are contributing to catalysis when visual inspection of the gels suggests that differences are small. In fact the mutations introduced seem to have a bigger impact on chain length, rather than overall catalytic activity. What is absent from the paper is determination of the effects of mutations on the activity of full length Ark2C when presented with a bona fide substrate. The lysine discharge assays are very nice at determining the ability of the RING construct to activate the Ub~E2 thioester, but autoubiquitination of the E3 and the production of unanchored ubiquitin chains are not a great substitute for a real substrate.

Reviewer #1 (Remarks to the Author):

This is a well-written, clearly communicated study, reporting the finding of elements within the Ark2C RING (and by conservation, Arkadia) that regulate ubiquitin transfer through interactions with catalytic and non-catalytic ubiquitin molecules. The Day lab has a well-regarded track record in understanding the finer details of ubiquitin molecules regulating ubiquitin transfer processes, and this work builds upon and extends earlier findings.

The main results are the determination of a complex of Ark2C RING in complex with E2-Ub, and also bound to another ubiquitin molecule by virtue of a fusion to the N-term of the Ark2C construct. The quality of the structural biology is high, and the findings are supported by careful biochemistry that validate the interactions between the proteins, and the role those interactions play in the transfer process.

In this reviewer's opinion, the data are sound, the conclusions are supported by the experimental results presented, it is not 'oversold' as being the fundamental way all E3 ligases work, rather it is carefully discussed in the context of the functional roles the ligases play. I found the figures easy to follow, and have no major concerns. I have one curiosity question which the authors may be able to address, which is given the propensity for Arkadia to bind SUMO and polySUMO chains, do they think that a fusion of SUMO instead of UbR might achieve the same stabilisation of the activated E2-Ub they observe in their structure?

Thank you, we agree that it is indeed intriguing to consider SUMO molecules as regulators of Ark2C. Although they have a highly similar structure, sequence conservation is low. Notably, none of the residues identified in our study as making important contacts are conserved in SUMO. When mapping the very few residues that are conserved onto the activated complex structure, it appears that these are optimised for the contact with RING and E2 resembling Ub^D in the closed conformation. Although we think SUMO is unlikely to activate Ark2C, it will be something to test in the future.

A few minor points -

p7. line 167 - end of sentence may need a specific reference to support the Lys11 statement.
Thank you, we have included reference 25 here.

p7. line 171 - should be burying 375 Å² not just Angstrom (missing a squared symbol)
Thank you, correction made.

p7. line 191 - and in materials and methods - when describing the S22R version of Ubch5 that cannot build polyubiquitin chains, there should be a reference to the original paper describing this - Hibbert and Sixma, J Biol Chem 2012 Nov 9;287(46):39216-23. doi: 10.1074/jbc.M112.389890.

Intrinsic flexibility of ubiquitin on proliferating cell nuclear antigen (PCNA) in translesion synthesis PMID: 22989887

Thank you, reference included at both points.

Reviewer #2 (Remarks to the Author):

Paluda et al. have solved the crystal structure of a ubiquitin Ark2C linear fusion in complex with an E2 ubiquitin conjugate. This follows a paper by Wright et al. in NSMB in 2016, where a model

of such a complex was proposed. In this structure the E2 ubiquitin conjugate is captured in the closed conformation indicating an active complex. As Ark2C is functional as a monomeric RING E3 ligase, the closed conformation of the E2 ubiquitin conjugate is stabilized by the linearly fused ubiquitin to Ark2C, which binds to the ubiquitin molecule conjugated to the E2. Biochemical analysis of mutations at this interface confirms the importance of this interaction for catalysis. The crosslinking mass spectrometry experiment reveals an interaction between the β 3-RING loop of Ark2C and ubiquitin. The authors also propose a model for K48 linked ubiquitin chain formation based on crystal packing.

Structural results confirm non-covalent ubiquitin as a modulator of RING E3 ligase activity and mass spec crosslinking exposed a previously unknown role for the β 3-RING loop. In the title of the manuscript, there are two elements being investigated: the effect of non-covalent ubiquitin and the effect of the β 3-RING loop of Ark2C on catalysis. However, the biochemical analysis presented in the paper is not sufficient to validate the importance of specific interactions predicted from the structure and does not clearly dissect out the separate roles of the regulatory ubiquitin and the β 3-RING loop on Ark2C catalysis.

Specific points:

1. Figure 1b and c: It is not clear in these figures and not mentioned in the main text on page 5 which residues of the Ubiquitin Ark2C linear fusion are resolved and not resolved in the crystal structure. There are multiple termini in Ark2C in Figure 1c, however, there are no labels on the figure to help the reader understand what they correspond to. The authors need to label the figure and specify which residues are resolved.

Thank you for the suggestion. The labelling of structural figures has been improved throughout and we have also included dashed lines to highlight the residues that are missing from the linker between UbR and Ark2C, as well as between the RING domain and N-terminal β -strand. In addition we have specified which residues have been resolved in the text.

2. Figure 1e: Ark2C alone (without ubiquitin linearly fused to it) is required as a control to show the effect of the absence of a non-covalent ubiquitin molecule. Considering Ark2C retains significant activity compared to the ubiquitin Ark2C linear fusion (Supplementary Figure 1c), it is therefore important that Ark2C alone is included in Figure 1e, to accurately determine the role of non-covalent ubiquitin in the mechanism of Ark2C.

We did not include this comparison in our original manuscript as we have previously reported data showing the impact of fusing ubiquitin to the RING domain of Ark2C on activity (Figure 3b, Wright et al., 2016). However, our published data was done using an oxyester Ub_{ch5b}~Ub conjugate and we have now included comparable experiments with the thioester conjugate in the supplementary data (Supplementary Figure 1d).

3. Figure 1f: Again, Ark2C alone is required to show the effect of mutations in the E2 ubiquitin conjugate on Ark2C catalysis in the absence of non covalent ubiquitin.

As for the previous point we have now added these assays to the supplementary data (Supplementary Figure 3e). This data shows that Ark2C alone is considerably delayed compared to the fusion and the mutants are comparable to Ark2C alone.

4. Supplementary Figure 1c: Both Ark2C and ubiquitin Ark2C are tested in this figure and both appear to show similar catalytic rates. However, in the main text (page 5, line 116) the authors claim that ubiquitin transfer is affected. The difference appears to be in the length of ubiquitin chains formed, with ubiquitin Ark2C favouring longer ubiquitin chains. It is difficult to determine a difference between the amount of free ubiquitin consumed in the assay as ubiquitin is overloaded and not quantified. Quantitation is required to determine the effect on ubiquitin transfer. In addition, the band for Ubiquitin Ark2C is stronger than Ark2C alone. This may be due to differences in staining, however, it is important to ensure E3 ligase concentration is the same when comparing their activity.

As the referee notes the UbArk2C fusion protein makes longer chains, but the data was not quantified. For these assays we do not think ubiquitin consumption is a good indicator of activity as converting all ubiquitin to diubiquitin or long chains will appear the same. However, if 10 molecules of ubiquitin form diubiquitin, 5 bonds are required, whereas a chain of 10 molecules will have 9 bonds.

Instead we now include data showing the assay with labelled ubiquitin and we have made a line scan plot for the 10 minute time points (Supp Figure 1c). In this case, the linescan shows both greater consumption of ubiquitin and greater chain building, and as the chains are very long it suggests that UbArk2C has considerably greater activity. We have also included a discharge assay for the same constructs which clearly highlights the difference in activity (Supp Figure 1d). We have also modified the text.

The referee also queries quantification of the E3s, we agree this is very important and we have taken considerable care using multiple approaches. As the referee notes the two proteins stain differently and this is why UbArk2C appears darker.

5. Figure 2a: It is not clear from the figure whether K6 or K48 is closer to the E2 active site. Distance measurements are required in this figure.

Distance measurements have been added, which show that K48 is closer.

6. Figure 2b: It is not clear in this assay that ubiquitin chain assembly was significantly reduced for the R125D mutant, as mentioned in the main text on page 7, line 174. Similar amounts of free ubiquitin are consumed and there is also an accumulation of even longer ubiquitin chains at the top of the gel for the R125D mutant. Quantitation is required to determine the effect on ubiquitin chain formation. In addition to K6 and K48 linked ubiquitin chains, K63 and K11 linked ubiquitin chains can be assembled by UbcH5b. To claim that this is a model for K48 linked ubiquitin chain formation, mass spec analysis of the mixture of chains needs to be performed on both WT and the R125D mixture to determine the type of ubiquitin chains formed and whether the R125D mutation alters the specificity.

We agree with the referee that the data related to this point could be strengthened. To address this, in the revised manuscript we have compared the ability of UbcH5b^{WT} and the R125D mutant to assemble chains when provided with Lys48-only ubiquitin. This shows a clear difference in activity and we have now included this as Figure 2b. The assay with WT ubiquitin has been moved to the supplementary data (Supplementary Figure 4c).

In addition we have modified the text in the results section to emphasise that these contacts are just one way that Lys48 can be positioned for chain formation.

7. Figure 3a: The S22R mutation, which perturbs backside binding of ubiquitin to UbcH5a, is more deleterious to longer ubiquitin chains formation than the M313A mutation, which perturbs UbR binding to Ark2C. Quantitation is required to determine the difference between WT and M313A and this finding should be reported and discussed in the main text.

These assays have been repeated and linescans included to more clearly show the difference in activity. We have also revised the text.

8. Figure 4 a and b: As mentioned for Figure 1b and c, the authors need to specify which residues of the Ubiquitin Ark2C linear fusion are resolved and which residues are not resolved in the main text (page 8, line 207).

Addressed as discussed above (point 1).

9. Supplementary Figure 5a: In the crosslinking mass spec experiment, the authors cannot distinguish between which ubiquitin molecule (UbR or UbD) the β 3-RING loop of Ark2C is crosslinking to. Comparing Ub-Ark2C and Ark2C in the cross-linking experiment would be useful in this respect, but the issue could be resolved by using, heavy isotope labeled UbD to set up the cross-linking analysis. This is an important point as differentiating between ubiquitin molecules is critical to the mechanism proposed.

Labelled forms of ubiquitin were not readily available to us, therefore to address the referees query we have used a zero-length cross-linking reagent, EDC, which crosslinks acidic and basic residues that are in close proximity. Using this cross-linker, the top Ub-Ark2C crosslinked hit was between Lys11 of ubiquitin and E287 of Ark2C, Lys11 also crosslinks to D289. Given the structure and the length of the crosslinker this result strongly supports the data shown in Figure 5b and we have now included this data as Figure 5c and 5d. We have also modified the text.

10. Figure 4c: Ark2C alone is used in this discharge assay and given that Figure 1e shows the UbD and UbR interaction is very important it is surprising that discharge activity is observed in WT Ark2C alone. The Ubiquitin Ark2C linear fusion is required for comparison in Figure 4c and quantitation of the results is required as per Figure 1e.

The referee identified an important point as the conditions of the assay differ. The difference (higher lysine concentration) is now explicitly stated in the methods and figure legend. We have also addressed this concern in two ways – first the activity of Ark2c and UbArk2c have been compared under identical conditions (Supplementary Fig. 1c and 1d). In addition we have quantified these discharge assays and this data is included in Figure 4c and 4g.

11. Supplementary Figure 5c: The authors claim that chain building is impeded by the delta basic and delta acidic forms of Ark2C in this figure, however, long ubiquitin chains are still formed by both forms of Ark2C. Similar amounts of free ubiquitin are consumed in the assay also. Quantitation of this assay is required to determine if ubiquitin transfer is impaired as well as the formation of different lengths of ubiquitin chains.

We agree that the differences are modest, but we believe that chain formation is impeded. We have modified the text to address this. The relevant section now reads 'Although in multi-turnover assays the deletion mutants only showed a modest reduction in activity (Supplementary Fig. 5c), in single turnover assays both deletion constructs had a considerably diminished ability to promote ubiquitin release when incubated with the UbCH5b thioester conjugate (Fig. 4c).'

12. Figure 5b and c: Removing the lysine side chain of K11, K29 and K33 in ubiquitin by mutating to alanine does not show any effect on activity in these figures. Considering only mutation to the opposite charge, either D or E, effects activity, this indicates that these residues are not contributing to specific interactions and mutating to opposite charge is resulting in repulsion. In addition, the fact that Ub KK/EE runs differently on the gel makes visual interpretation of the gel particularly difficult. As a result, the biochemical analyses to support the hypothesis that the acidic residues of the β 3-RING loop of Ark2C interact with UbD are weak.

We thank the reviewer for these comments. We respond in several ways. First, we agree that there is only a modest reduction in activity with the ubiquitin Ala mutants and that these mutants suggest that charge contacts are important. Prompted by point #9 above we completed an EDC crosslinking experiment. This showed a strong crosslink between Lys11 in ubiquitin and several acidic residues (Glu287 and Asp289) at the end of the β 3-RING linker. Given the structure and mutagenesis data, this crosslink provides compelling evidence to support the model proposed, whereby Lys11 of Ub^D is somewhat flexible but makes contacts with acidic residues in the loop of Ark2C.

The referee also queries the validity of the charge complementation data. This type of analysis is quite common when analysing protein interactions [e.g. Figure 6 in Ge et al., (2018) PNAS 115, 4649], with the individual mutants predicted to disrupt an interaction due to repulsion, but when the charges are swapped and the two mutants combined, a recovery in activity, as seen here, is predicted.

While we appreciate the referees concerns, we feel that this data strongly supports the model proposed. Furthermore, the model is consistent with those that have been recently reported for TRIM21 and MDM2 (Magnussen *et al.*, Nature Communications 2020 and Kiss *et al.*, Nature Communications 2019).

In summary, the authors have determined an interesting structure which reveals the role of a bound regulatory ubiquitin and suggests a role for the loop between b3 and the RING that is not resolved in the structure. Further biochemical and mass spec experiments will be required to validate the role of the regulatory ubiquitin and determine the key interactions made by the b3-RING loop. Apart from the assays shown in Fig 1e, f the biochemical assays have not been subjected to quantitative analysis. This is important as the authors are making claims that particular residues are contributing to catalysis when visual inspection of the gels suggests that differences are small. In fact the mutations introduced seem to have a bigger impact on chain length, rather than overall catalytic activity. What is absent from the paper is determination of the effects of mutations on the activity of full length Ark2C when presented with a bona fide substrate. The lysine discharge assays are very nice at determining the ability of the RING construct to activate the Ub~E2 thioester, but autoubiquitination of the E3 and the production of unanchored ubiquitin chains are not a great substitute for a real substrate.

We thank the referee for their careful review of our manuscript. We have now done a number of additional assays and quantified many of these, with the new data strongly supporting the original conclusions. In addition, we have added new cross-linking data that shows the interactions previously proposed occur in solution. We agree with the referee that the next step is to investigate substrate ubiquitylation. That analysis is beyond the scope of this manuscript, but we agree that it will be very interesting to do so in the future.

REVIEWERS' COMMENTS

Reviewer #1 (Remarks to the Author):

In my opinion in this revised article all the previous concerns raised have been adequately addressed.

Reviewer #2 (Remarks to the Author):

I think that the additional biochemical experiments and their quantitation have strengthened the manuscript. The new crosslinking experiments are a nice addition to the paper. Given that the authors have addressed our original queries I am now supportive of publication.